# Brain Activations and Functional Connectivity Patterns Associated with Insight-Based and Analytical Anagram Solving

**DOI:** 10.3390/bs10110170

**Published:** 2020-11-08

**Authors:** Dmitry O. Sinitsyn, Ilya S. Bakulin, Alexandra G. Poydasheva, Liudmila A. Legostaeva, Elena I. Kremneva, Dmitry Yu. Lagoda, Andrey Yu. Chernyavskiy, Alexey A. Medyntsev, Natalia A. Suponeva, Michael A. Piradov

**Affiliations:** 1Research Center of Neurology, 125367 Moscow, Russia; d_sinitsyn@mail.ru (D.O.S.); bakulinilya@gmail.com (I.S.B.); milalegostaeva@gmail.com (L.A.L.); moomin10j@mail.ru (E.I.K.); dmitrylagoda.doc@gmail.com (D.Y.L.); andrey.chernyavskiy@gmail.com (A.Y.C.); nasu2709@mail.ru (N.A.S.); mpi711@gmail.com (M.A.P.); 2Valiev Institute of Physics and Technology, Russian Academy of Sciences, 117218 Moscow, Russia; 3Institute of Psychology, Russian Academy of Sciences, 129366 Moscow, Russia; medintseff@yandex.ru

**Keywords:** insight, Aha!-moment, fMRI, resting-state fMRI, functional connectivity, anagrams, creativity

## Abstract

Insight is one of the most mysterious problem-solving phenomena involving the sudden emergence of a solution, often preceded by long unproductive attempts to find it. This seemingly unexplainable generation of the answer, together with the role attributed to insight in the advancement of science, technology and culture, stimulate active research interest in discovering its neuronal underpinnings. The present study employs functional Magnetic resonance imaging (fMRI) to probe and compare the brain activations occurring in the course of solving anagrams by insight or analytically, as judged by the subjects. A number of regions were activated in both strategies, including the left premotor cortex, left claustrum, and bilateral clusters in the precuneus and middle temporal gyrus. The activated areas span the majority of the clusters reported in a recent meta-analysis of insight-related fMRI studies. At the same time, the activation patterns were very similar between the insight and analytical solutions, with the only difference in the right sensorimotor region probably explainable by subject motion related to the study design. Additionally, we applied resting-state fMRI to study functional connectivity patterns correlated with the individual frequency of insight anagram solutions. Significant correlations were found for the seed-based connectivity of areas in the left premotor cortex, left claustrum, and left frontal eye field. The results stress the need for optimizing insight paradigms with respect to the accuracy and reliability of the subjective insight/analytical solution classification. Furthermore, the short-lived nature of the insight phenomenon makes it difficult to capture the associated neural events with the current experimental techniques and motivates complementing such studies by the investigation of the structural and functional brain features related to the individual differences in the frequency of insight-based decisions.

## 1. Introduction

The study of various aspects of problem solving and creativity is an intensively developing area of studies in modern cognitive science [1,2,3,4,5]. One of the most mysterious problem-solving phenomena is insight, the sudden emergence of a solution, often accompanied by the subjective Aha! Experience (Eureka) [6,7,8,9,10,11,12]. The definitions of insight in the literature include several features [6,8,12]: (1) insight manifests as a sudden and unexpected discovery of the correct solution; (2) a key feature of insight is that the subject has no access to the steps leading to the solution; (3) insight is often (but not always) accompanied by a burst of positive emotions; (4) insight is often preceded by long unproductive attempts to find a solution. Insight is contrasted with analytical problem solving which is conscious and advances step by step so that the subject can explain how they reached a solution [12]. Examples of insight solutions to a variety of problems are widely represented in both professional activities and everyday life [8,10,11,12]. This determines the significant interest in studying the phenomenon of insight to improve the understanding of the neural bases of creativity.

Various tasks are used to study insight. Most widespread are the so-called “insight tasks”, such as the “Nine-dot problem” [13]. Solving these problems requires restructuring their representations and abandoning the usual algorithms or solutions, thus removing the functional fixedness [14]. Despite certain advantages, these tasks have a number of limitations that make them difficult to employ in studies using modern neurophysiological and neuroimaging methods. In particular, the solution of these problems usually takes considerable time; furthermore, the tasks themselves are rather complex and relatively rarely solved correctly by the subjects. In addition, each task can be presented for solving a problem only once [12].

In view of these limitations of classical insight problems, several other task types were proposed. Their key aspects are the possibility of creating a virtually unlimited number of tasks of the same type and similar complexity, as well as a high incidence of correct solutions in a relatively short time period. In addition, such problems can be solved both by insight and analytically, the choice of which is subjectively determined by the solver. Examples of such tasks are Mednick’s Remote Associates Task (RAT), CRA (Compound Remote Associations), the matchstick arithmetic problems, Chinese logogriphs, rebus puzzles, riddles, anagrams, etc. For example, in solving an anagram, the presented letters must be rearranged to obtain a word (e.g., NHTGISI = INSIGHT) [8,12,15]. 

These tasks are widely used to study the neuronal bases of insight using neurophysiological and neuroimaging techniques, in particular, functional MRI (fMRI), electroencephalography (EEG), and evoked potentials [8,12,16]. Most of these studies analyzed the correlation between the activation of various brain structures and the emergence of an insight-based solution to a problem. The studies comparing the patterns of brain activation during insight and analytical solving of the same task may be of particular interest. For example, the use of functional MRI and EEG showed that insight problem solving, compared with the analytical strategy, was associated with an activation in the temporal cortex of the non-dominant hemisphere [17]. Another study reported that insight anagram solving, compared with the analytical approach, was accompanied by a significant activation of the bilateral insulae, Broca’s area, right prefrontal cortex, and anterior cingulate cortex [18].

A meta-analysis of 13 fMRI studies revealed 11 activation clusters associated with insight problem solving and located in several brain regions including the left premotor and supplementary motor cortices, left and right middle temporal gyri, left and right precunei, left cingulate gyrus, right superior frontal gyrus, left claustrum, right and left insulae, and other structures [12]. The involved brain regions (the insight network) have a degree of overlap with salience, executive control, fluid intelligence, and creativity networks [12]. It should be noted that the analysis of the research findings is hindered by the use of different tasks and experimental paradigms, as well as different approaches to insight detection. One of the limitations of insight studies using functional MRI is the small sample size in the majority of the papers (7 to 30 participants in the studies included in the meta-analysis, with 11 out of the 13 papers investigating fewer than 20 subjects).

Another direction of current investigation is the use of neuroimaging methods for finding the basic structural and functional features of the brain underlying individual differences in the frequency of insight solutions [8]. Resting state fMRI (rs-fMRI) is one of the common methods in such studies, allowing the assessment of the functional connectivity (FC) between various brain regions [19,20]. Currently rs-fMRI is widely used in cognitive science, basic and clinical brain research [21]. Spontaneous brain activity can play an important role in mind wandering, mental simulation, and divergent thinking in general [3]. At the same time, the relationship between FC and spontaneous brain activity to the insight phenomenon remains poorly understood. Two studies demonstrated the possibility of detecting the correlation between the FC of some brain structures and the insight frequency [22,23]. These data complement the results of the studies using EEG recordings [24,25,26] and, in general, suggest the possibility to identify basic features of resting brain activity that are associated with insight. Further development of this line of research seems relevant and promising. The application of rs-fMRI and task fMRI in a single study may provide useful information regarding the structures underlying insight problem solving. 

In our research, we used task fMRI and resting-state fMRI to study the neural basis of insight. We analyzed the regions of the brain that are preferentially activated when solving anagrams by using insight compared to an analytical strategy, as well as the activations during each type of anagram solving compared to rest. In addition, we analyzed the correlation of functional connectivity probed by rs-fMRI with the frequency of insight solutions of anagrams and the frequency of correct solutions. 

## 2. Materials and Methods

### 2.1. Subjects and fMRI Acquisition Parameters

All procedures performed in the study were in accordance with the ethical standards of the Ethics Committee of the Research Center of Neurology (reference number 2-4/19 from 20.02.2019) and with the 1964 Helsinki declaration and its later amendments or comparable ethical standards. Written informed consent was obtained from all the participants.

Inclusion criteria for participation in this study:age of 18–55 years;normal or corrected-to-normal vision;Exclusion criteria;MRI contraindications, e.g., implanted cardiac devices, claustrophobia, pregnancy, and others according to the guidelines [27];intake of drugs acting on the central nervous system;history of neurologic or psychiatric disorders;severe chronic diseases.

Healthy subjects were preselected in a preliminary test involving tasks similar to those used in the fMRI experiment (see Section 2.2). A subject was included in the main study if two conditions were met as a result of the preliminary test: 1) the number of solved anagrams was greater than or equal to 50% of all the anagrams presented; 2) the number of insight solutions was greater than or equal to 40% of all the correct solutions. Similarly to the main experiment, we used anagrams with a known average rate of solvability [28]. The anagrams consisted of 5 to 8 letters, the solvability rate ranged from 40% to 60%, and the average solution time was 8 seconds. In total, the subjects were presented with 40 anagrams divided into series of 10 anagrams each. Fifty healthy volunteers participated in the preliminary test, and 32 of them were selected for the main study (20 women, age median 20.5, age quartiles 19.0 and 24.5).

The MRI acquisitions were performed on a 3T Siemens MAGNETOM Verio clinical scanner (Siemens, Erlangen, Germany). Anatomic images for spatial normalization were obtained using a 3D-T1-gradient echo sequence (T1-MPR) and consisted of 176 sagittal slices (repetition time (TR) = 1900 ms, echo time (TE) = 2.47 ms, slice thickness = 1.0 mm, voxel size 1.0 × 0.977 × 0.977 mm^3^, field of view (FOV) = 250 mm). Task fMRI data were acquired using a T2*-gradient echo sequence (TR = 2 s, TE = 21 ms, slice thickness = 3 mm, voxel size = 3.0 × 3.0 × 3.75 mm^3^, FOV = 192 mm, number of slices = 36). The first 5 functional images were discarded to achieve dynamic magnetic equilibrium. The resting state fMRI data for each subject contained 190 scans with the parameters: TR = 2.4 s, TE = 30 ms, slice thickness = 3 mm, voxel size = 3.0 × 3.0 × 3.75 mm^3^, FOV = 192 mm, number of slices = 36.

### 2.2. fMRI Paradigm

Before the experiment, the participants received the following definition of an insight solution to be used in the self-evaluation of each trial: “An insight solution is a solution of an anagram that came to you unexpectedly, and you cannot give a subjective report on how it came to you. If you can explain how you reached a solution (step by step), it is a non-insight one. If in doubt, press ‘no insight’”.

The task involved solving Russian-language anagrams with 5–8 letters. The stimuli were presented on a display through a folding mirror placed on the head coil. The subjects provided their responses through hand-held button pads in both hands. The fMRI experiment started with instructions to press each of the buttons to ensure the subject knew their names (by which they were referred to in the response options shown at every stage of the paradigm). This was followed by one training session of solving 5 anagrams (not included in the analysis) and 5 sessions each containing 10 anagrams, with one-minute breaks between the sessions. The structure of a single stimulus presentation is shown in Figure 1. It involved up to three presentations of the anagram for 4, 8, and 8 seconds respectively, interspersed with two questions asking if the subject was close to a solution. In case the subject pressed the button with the meaning “solved” during any anagram presentation interval, the subject would be asked if the solution came through insight, which was followed by two questions about the first and last letters of the subject’s answer (aimed at verifying the solution), and the display of the correct answer. If the subject had not pressed “solved” in any of the anagram presentation intervals, the correct answer was shown, after which time the next trial started.

### 2.3. Data Analysis

The task fMRI data analysis was performed using SPM12 (Functional Imaging Laboratory, Wellcome Department of Imaging Neuroscience, Institute of Neurology, London, UK) and MATLAB R2017a (Mathworks, Natick, MA, USA). The preprocessing consisted of the following steps: slice timing correction, realignment of functional images, coregistration of anatomic and functional data, normalization to the standard Montreal Neurological Institute (MNI) template, and spatial smoothing of the functional scans with a Gaussian kernel of 8 mm FWHM. Three subjects were excluded due to high-amplitude head motion (shifts exceeding 5 mm or rotations greater than 4 degrees).

The fMRI data was described by using a general linear model with regressors obtained by convolution of boxcar functions corresponding to the modelled neural states and events with the canonical hemodynamic response function. Regressors determined by head movement parameters were also taken into account. As the time intervals during which the neural activity is supposed to occur are associated with the solution of an anagram, we selected all the intervals in which the anagram was displayed on the screen. For each anagram presentation interval, the fMRI signal was analyzed in the time window from its beginning to 20 s after its end (a latency at which the hemodynamic response function is considerably attenuated). The analysis also included periods of rest between the series of anagrams. All other time intervals were excluded from the consideration by adding regressors for individual scans in order to reduce the impact on the model of the activity associated with answering the auxiliary questions. The following conditions were simulated: the subject’s work on the task in the case of a subsequent correct insight solution, similar states for the trials with a correct analytical solution, without a solution, or with the wrong solution, the state of rest between the series of tasks, the state of the subject’s thinking on questions about the closeness to a solution, the solution type (insight or analytical), and the first and last letters of the answer (validation), as well as viewing the correct answer. Additionally, three types of events (with zero duration) were used to simulate the pressing of each of the three buttons.

For each subject, the model parameters were estimated using the fMRI data, after which the images corresponding to the following contrasts were calculated: insight solution–rest, analytical solution–rest, insight solution–analytical solution. The obtained images were analyzed by using the method of summary statistics for a random-effects group-level model. To assess the statistical significance of activations in the obtained group-level parametric maps, one-sided tests were used with a cluster correction (based on the theory of Gaussian random fields, as implemented in SPM) controlling the familywise error rate (FWER) at the level of 0.05 for a voxel-level feature-inducing threshold corresponding to *p* = 0.001.

To assess the relationship of the results to previous findings, we compared the activations to a set of 11 areas determined in the meta-analysis [12] by using an activation likelihood estimation (ALE) meta-analysis of 13 fMRI studies using several different insight tasks. Additionally, for each contrast, we computed its values averaged over the voxels in every one of the 11 areas, and tested them for significance with a control of false discovery rate (FDR) at 0.05. For this, we used the MarsBaR toolbox (version 0.44, http://marsbar.sourceforge.net/) [29].

The resting-state fMRI data was processed using the CONN functional connectivity toolbox [30], version 18a, and SPM12. The preprocessing consisted of the following steps: segmentation of structural data, realignment of functional images (motion correction), slice timing correction, coregistration, normalization into the MNI space, outlier detection/scrubbing using the artifact detection tool (ART), and spatial smoothing with a Gaussian kernel having a FWHM of 8 mm. Denoising was performed by removing the following confounders by using linear regression: (1) the blood-oxygen-level-dependent (BOLD) signal from the white matter and cerebrospinal fluid (CSF) masks (5 principal components of each signal); (2) scrubbing (as many regressors as identified invalid scans); (3) motion regression (12 regressors: 6 motion parameters + 6 first-order temporal derivatives). The resulting signals were band-pass filtered in the range 0.008 to 0.09 Hz. 

Functional connectivity was studied under the region of interest (ROI)-to-ROI and seed-to-voxel approaches using the 11 ROIs from the meta-analysis of fMRI studies of insight [12]. Fisher-transformed correlation coefficients of the denoised ROI signals with each other and with all the voxels in the brain were studied for correlation with the insight rate (i.e. the fraction of the correct anagram solutions that were obtained by insight) and the number of solved anagrams. Two-sided tests were used to study both positive and negative correlations between the connectivity and the behavioral measures. Statistical significance of the ROI-to-ROI results was tested at an analysis-level FDR of less than 0.05. The seed-to-voxel maps were thresholded using the Gaussian random field theory with the cluster-wise FWER controlled at 0.05 for a voxel-level feature-inducing threshold corresponding to *p* = 0.002.

## 3. Results

### 3.1. Behavioral Statistics of Insight-Based and Analytical Anagram Solving

The fraction of the anagrams correctly solved by the participants was 0.6 (0.48, 0.72) (median and quartiles, here and below). Among the correct solutions, those obtained by insight constituted 0.66 (0.56, 0.79). The median time spent solving an anagram by insight was 5.7 s (4.9 s, 6.6 s), and for the analytical solutions, 6.2 s (7.9 s, 10.8 s). The insight-based solutions were faster than the analytical ones (Wilcoxon *p* < 0.001, *Z* = −4.11, signed-rank statistic: 44, median of the time differences 2.5 s).

### 3.2. Brain Activations During Insight-Based and Analytical Anagram Solving

In the contrast between the insight solutions and rest, a set of significantly activated regions was found (Figure 2), with the largest clusters located in the left premotor and primary motor cortices, as well as bilaterally in the occipital and parietal lobes (Table 1). The comparison of these activations with the 11 areas from the meta-analysis [12] (Table 2) showed considerable overlaps (Figure 3). The analysis of average contrast values in these areas showed significant effects in 8 out of the 11 ROIs (Table 3).

In the analysis of brain regions that were more active during analytical anagram solving than during rest, a pattern similar to the previous one was found (Figure 4, Table 1). The tests of the mean effects in the areas from the meta-analysis [12] showed that 9 out of the 11 areas were significantly activated (Table 3).

In the contrast between the insight and analytical anagram solving, there was one significant cluster of activation located in the right precentral and postcentral gyri (Figure 5, Table 1). The 11 areas from the meta-analysis did not show significant effects in this contrast (Table 3). 

### 3.3. Functional Connectivity Associated with the Rate of Insight Solutions

The ROI-to-ROI analysis of functional connectivity between the areas from the meta-analysis [12] did not reveal any significant correlations with the insight rate or the number of correct solutions at an analysis-level FDR of less than 0.05.

In the seed-to-voxel analysis, for three of the 11 areas, the functional connectivity had significant correlations with the insight rate (Table 4). Positive correlations were found for the connectivity of the seed area #1 located in the left premotor cortex (Figure 6A) with three regions in the right superior parietal lobule (FWER = 0.002), right superior frontal gyrus (FWER = 0.02), and right postcentral gyrus (FWER = 0.03) (Figure 6B). There was also a significant positive correlation of the insight rate with the connectivity between the seed area #4 in the left claustrum (Figure 7A) and an area in the right superior and middle frontal gyri (FWER = 0.04) (Figure 7B). Additionally, a negative correlation was found for the connectivity between the seed area #7 in the left frontal eye field (Figure 8A) and an area in the left lateral occipital cortex (FWER = 0.009) (Figure 8B). Among these correlations, only the first one (between the left premotor cortex and right superior parietal lobule) survived a Bonferroni correction for the 11 tested seed areas. The analysis of correlations of the seed-to-voxel connectivity values with the number of correctly solved anagrams revealed no results surviving this correction.

## 4. Discussion

In this study, task fMRI revealed bilateral activation of a number of brain regions during anagram solving. There was a significant overlap between the activated regions and the areas identified in the meta-analysis of 13 insight-related fMRI studies [12]. It is important to note that in the present study, the identified activation patterns were revealed during both insight-based and analytical anagram solving as compared to the resting state. Differences in activation during insight and analytical solutions were found only for the right precentral gyrus (which may be related to the study design where the left hand was used to press a button in case of insight). At the same time, rs-fMRI analysis revealed positive correlations of the insight frequency with the seed-based connectivity of two areas (left premotor cortex and left claustrum) and negative correlation with the connectivity of one area (left frontal eye field) from the meta-analysis by Sprugnoli et al. [12], with the first of these correlations surviving the Bonferroni correction for the 11 tested seed regions. The frequency of correct solutions did not show correlations with functional connectivity surviving the correction. These results are discussed in more detail below.

One of the important findings of this study is the significant overlap (almost complete coincidence) of the areas activated during insight and analytical anagram solving. This may indicate the difficulty and insufficient accuracy of the distinction between insight and analytical solutions performed by the subjects, although the significantly shorter duration of insight trials compared to analytical ones suggests that there was objective difference between the two strategies. Several studies showed that the subjective “Aha! experience” does not always accompany the insight solving, and moreover may also be present in case of false insight [31,32]. Although several approaches to distinguish insight and analytical strategies have been proposed so far [13,33,34], the self-report approach is probably optimal [34] and is used in the majority of neuroimaging studies. This is partly explained by the fact that an alternative approach based on the “feelings-of-warmth measure” (subjective proximity to the solution) requires a rather long time for its full implementation. Furthermore, in some cases insight can be detected even if there is a sense of a gradual approach to the correct answer [34]. Hedne et al. showed that there was no difference between the insight and noninsight problem solving in warmth ratings but insights were characterized by higher accuracy and higher confidence ratings [35]. However, the subjectivity of determining the problem-solving strategy remains one of the important limitations of our study. 

The areas activated during both insight and analytical anagram solving largely overlapped with the significant clusters identified in the meta-analysis of 13 fMRI studies [12] involving various insight-related tasks and contrasts. While some of the included studies directly targeted the differences between insight and analytical solutions (e.g. [17,18]), others contrasted problems differing in some aspects supposed to affect the strategy but not uniquely dictating it (e.g., unconstrained and constrained problems in the study by Vartanian and Goel [36], or familiar and unfamiliar chunks in the paradigm used by Wu, Knoblich, and Luo [37]). Considering this fact, it can be hypothesized that most of the activation areas that we identified are associated with the process of solving the task and not with insight per se. This is also indirectly suggested by the overlap of the identified activation areas with the brain regions involved in divergent thinking, general cognitive control and attention, and belonging to several brain networks (default mode network (DMN), salience, executive control, and creativity networks).

Among the identified activated areas, the left premotor cortex is of particular interest. The significance of this brain region is indirectly confirmed by the data from both rs-fMRI (statistically significant correlation between the insight frequency and the functional connectivity of the left premotor cortex with three other regions) and task fMRI (statistically significant activation in the contrast between insight and rest). Our findings are consistent with the results of the meta-analysis of insight-related fMRI studies, in which the largest activation cluster was identified in the left premotor / supplementary motor cortex [12]. In addition, a number of studies have shown the role of the premotor cortex in performing tasks assessing divergent thinking and creativity [38,39,40,41]. The premotor cortex is part of the executive control network and is involved in planning and selection of new actions [42,43]. The inferior frontal junction (IFJ, Brodmann area 44/6) can play an important role in providing flexible cognitive control over information retrieved from memory and in activating task representation [41,44]. It is interesting to note that according to Zhu et al. [45], the right premotor cortex is the only portion of the cerebral cortex where the volume of the gray matter is significantly correlated with creativity indices according to the Creative Behavioral Inventory (CBI). Overall, our findings and the results of previous studies suggest an important role of the premotor cortex in insight and creativity events in general.

Additionally, we obtained data on the possible role of the left claustrum in insight anagram solving, which is consistent with the results of the meta-analysis [12]. The claustrum plays an important role in regulating the level of consciousness, as well as in integrating the areas of the cortex involved in maintaining voluntary attention [46,47,48]. Under experimental conditions, the claustrum supports cognitive control for optimal behavior in challenging cognitive settings [49]. According to studies in rodents, the claustrum is a component of the homologue of the default mode network and has connections with structures identified in the rodent homolog of the salience network [50,51], with both networks having a possible association with insight [12]. According to Sprugnoli et al. [12], at a moment of insight problem solving, the claustrum can act as a monitoring center during “mind-wandering”, allowing the “drift” of the mind during the first reorganization of the available information and the emergence of the correct solution from its subconscious representation. 

The right and left precunei were activated during both insight and analytical anagram solving, and their involvement was found in a number of previous studies [52,53] and the meta-analysis [12]. The precuneus is one of the DMN hubs, which has been shown to participate in divergent thinking in studies using various methods of investigation [23,54,55,56]. According to W. Ogawa et al. [23], insight test battery score significantly positively correlates with gray matter volume (GMV) in the right insula and middle cingulate cortex/precuneus. The activation of the insula during anagram solving, which was also previously identified in [18], may be related to the participation of this region in interoception and ensuring a balance between internal and external stimuli [57]. The middle temporal gyrus can play a role related to imagination, language and memory, and can promote restructuring of the available information [12]. In addition, its activation during creativity tasks has been shown in [2]. The left frontal eye field and its connection to the occipital cortex may be important for visually guided behavior, control of spatial attention, and visual cognition [12,58].

We have not obtained data on the role of the anterior cingulate gyrus in the process of solving anagrams and the emergence of insight, which is discussed in several studies. Furthermore, consistent with the results of the meta-analysis [12], our findings do not support the concept of the dominance of the right (non-dominant) hemisphere during insight problem solving. 

In the contrast between insight and analytical solutions, we identified only one cluster located in the right precentral gyrus, which was more active during an insight solution than an analytical one. Although we cannot exclude a role of the primary motor cortex in the emergence of insight [23] and creativity in general [59], in our view, a more obvious explanation for this finding lies in the details of the study design. The subjects pressed a button with the left hand when defining the solution as an insight (and with the right hand for analytical solutions). Thus, there was a consistent difference between insight and analytical trials in the motor activity occurring within a few seconds after the anagram solving time interval, which may explain the observed contrast in the primary motor cortex.

The article [18] used a similar task based on solving anagrams. One significant difference was that there was a 10-second rest interval after the button press upon solving an anagram. This may have helped to avoid the overlap of BOLD signal fluctuations related to solving the problem and the subsequent answers to additional questions, which could have increased the sensitivity of the analysis. On the other hand, this delay reduces the number of stimuli that can be presented in a given time interval, which decreases the power of the tests. Thus, the value of introducing a rest interval requires verification. Meanwhile, the absence of a theoretically-grounded correction for multiple comparisons in the mentioned study makes it difficult to directly compare the results with those obtained in the present research.

It should be noted that, similarly to the present work, in [18], there was a difference in decision time between insight and analytical solutions, and it was considerably more pronounced than in our data (analytical solutions lasting more than three times longer than insight ones). This fact raises the question of whether the activation amplitudes of brain regions are different for solutions of the same type, but of different duration, and whether this effect can explain the observed differences between insight and analytical solutions (for example, short high-amplitude activation during quick solutions, including most of the insights, and longer-lasting, lower-amplitude activation of the same regions for slow solutions, including most of the analytical ones, which is consistent with the activation profiles for some areas in [18]). Developing an appropriate control procedure for this aspect may be an important area for further research.

This study has several limitations, one of them being the small sample size. Other important limitations consist in the problems described above and are related to the subjectivity of the subjects’ determination of the problem-solving strategy (insight or analytical solution), as well as the pressing of the buttons with different hands after insight and analytical solutions. The proximity in time of the anagram solving intervals and the answers to auxiliary questions could have led to overlaps of the corresponding BOLD signal oscillations, reducing the sensitivity of the analysis. It is also worth noting that the reported median insight frequency estimate was obtained from a set of preselected subjects (Section 2.1) and may be biased upwards compared to the general population.

In our view, the findings suggest that insight studies can benefit from combining various neuroimaging techniques. Insight is likely to be an extremely short-lived (fleeting) event, and so its study using fMRI is limited by the low temporal resolution of this method. The use of evoked potentials and EEG for this purpose is limited by their low spatial resolution. In this situation, a possible approach is to search not only for specific activity occurring at the moment of insight, but also for persistent structural and functional correlates of individual insight occurrence rates in studies using rs-fMRI [22,23], MR-morphometry [23], resting-state EEG [24,25,26], and other techniques. Such studies can help identify differences in the structural and functional brain organization in subjects with varying frequency of insight solutions and different creativity scores. Furthermore, a promising area is the evaluation of changes in insight frequency upon modulating the activity of identified brain regions of interest in causal studies using non-invasive brain stimulation methods (see, e.g., [60,61,62]). 

## Figures and Tables

**Figure 1 behavsci-10-00170-f001:**
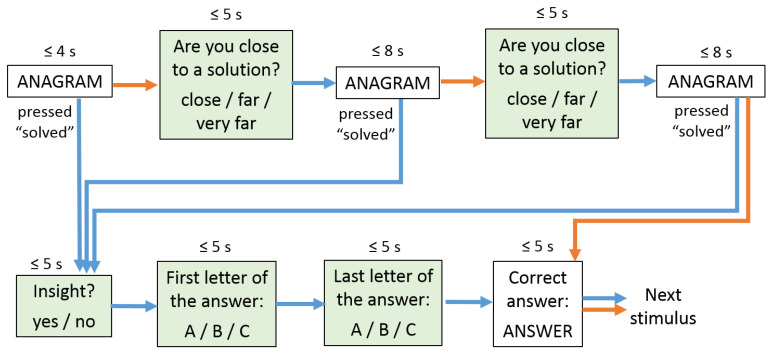
fMRI paradigm for insight-based and analytical anagram solving. The time indicated above each state (e.g., “≤ 4 s”) means that the state lasts until a button is pressed or until the specified time elapses, whichever happens first. Blue and orange arrows correspond to transitions triggered, respectively, by a button press and by elapsed time. For the states shaded green, an answer (by a button press) is required, and so if it is not given within the specified time, the current trial is interrupted with a corresponding message, and the next stimulus is presented.

**Figure 2 behavsci-10-00170-f002:**
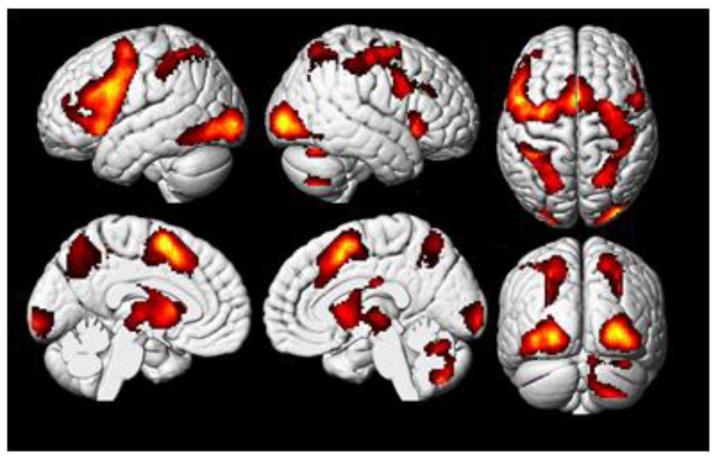
Regions of the brain with significant activation in the contrast between insight-based anagram solving and rest (“insight > rest”, cluster-wise familywise error rate (FWER) < 0.05).

**Figure 3 behavsci-10-00170-f003:**
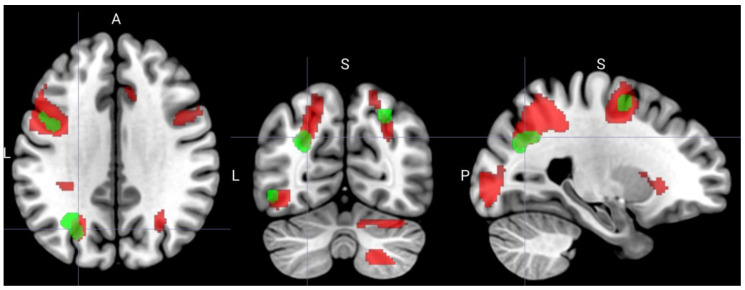
Comparison of the activations in the “insight > rest” contrast (red) with the set of 11 regions of interest (ROIs) (green) from the meta-analysis of functional Magnetic resonance imaging (fMRI) studies of insight [12].

**Figure 4 behavsci-10-00170-f004:**
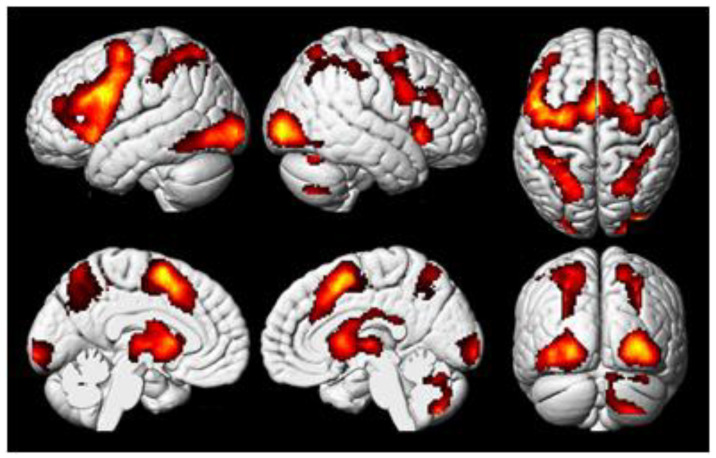
Regions of the brain with significant activation in the contrast between analytical (non-insight) anagram solving and rest (“analytical > rest”, cluster-wise FWER < 0.05).

**Figure 5 behavsci-10-00170-f005:**
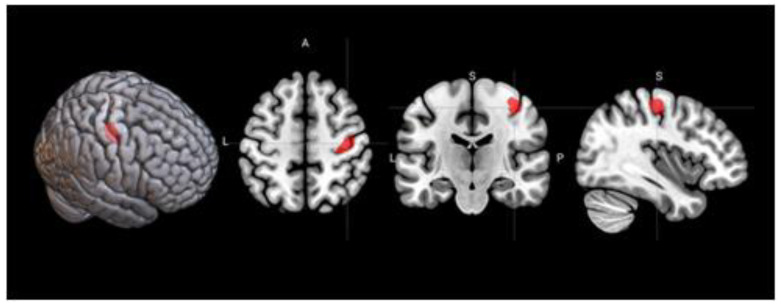
Region of the brain with a significant activation in the contrast between insight-based and analytical anagram solving (“insight > analytical”, cluster-wise FWER = 0.02).

**Figure 6 behavsci-10-00170-f006:**
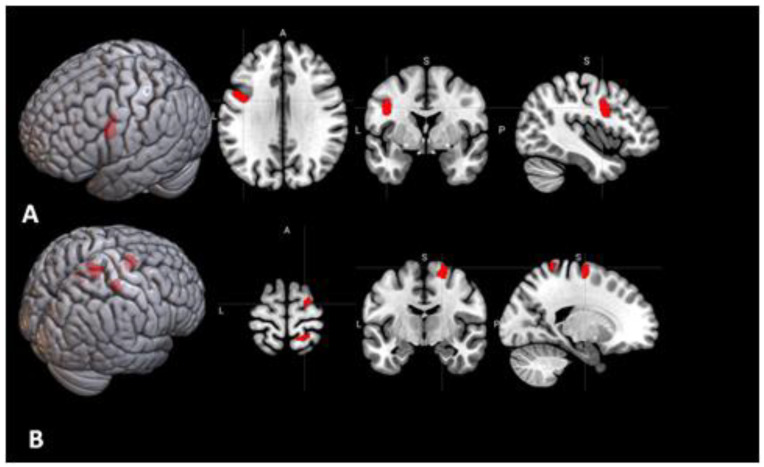
(**A**) Seed region #1 from the meta-analysis [12] in the left premotor cortex. (**B**) Areas of the brain for which the connectivity with the seed #1 shown in A had significant positive correlation with the insight rate (the fraction of the correct anagram solutions that were obtained by insight), thresholded at cluster-wise FWER < 0.05. L—left, A—anterior, P—posterior, S—superior.

**Figure 7 behavsci-10-00170-f007:**
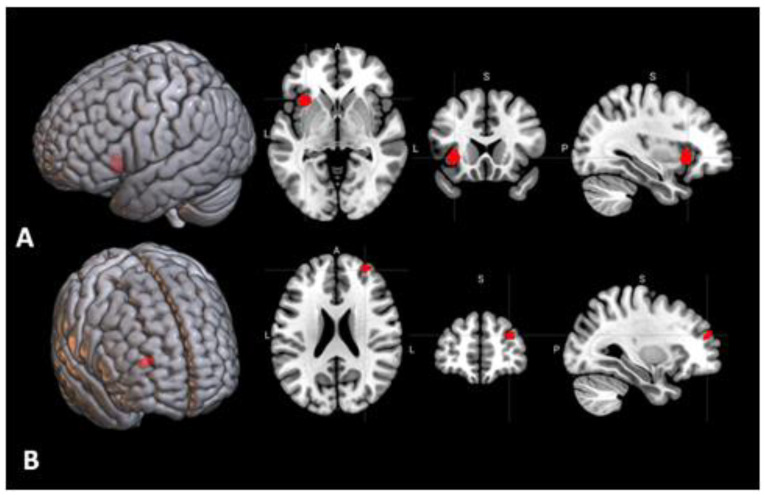
(**A**) Seed region #4 from the meta-analysis [12] in the left claustrum. (**B**) Area of the brain for which the connectivity with the seed #4 shown in A had significant positive correlation with the insight rate (FWER = 0.04).

**Figure 8 behavsci-10-00170-f008:**
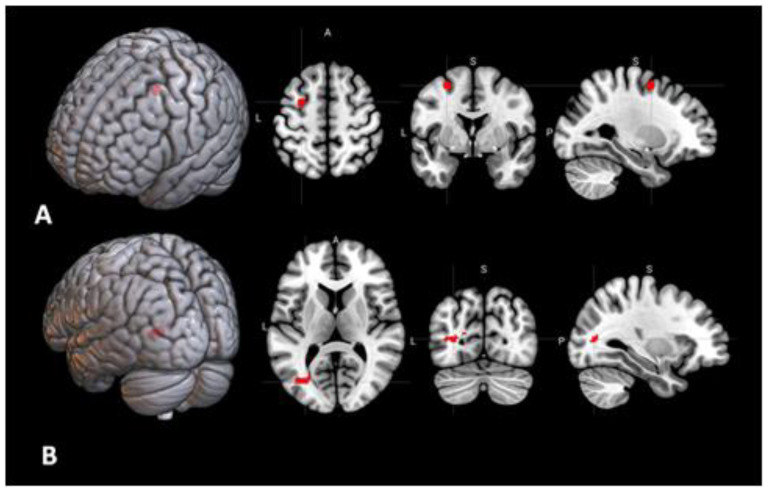
(**A**) Seed region #7 from the meta-analysis [12] in the left frontal eye field. (**B**) Area of the brain for which the connectivity with the seed #7 shown in A had significant negative correlation with the insight rate (FWER = 0.009).

**Table 1 behavsci-10-00170-t001:** Activation areas for the three contrasts considered. One-sided *t*-tests were used with a cluster correction controlling the FWER at 0.05 for a voxel-level feature-inducing threshold corresponding to *p* = 0.001. The voxel size of the normalized images was 2 × 2 × 2 mm^3^.

Gyrus/Region of the Peak	Peak MNI Coordinates (mm)			Peak *t* (28)	Number of Voxels	*p* (FWE, Cluster Level)
	*x*	*y*	*z*			
Contrast “insight > rest”						
Left inferior occipital gyrus	−26	−96	−2	14.78	2235	<0.001
Right inferior occipital gyrus	34	−94	0	12.97	1456	<0.001
Left precentral gyrus	−44	6	30	12.83	15,446	<0.001
Right cerebellum exterior	24	*−66*	−50	11.09	1085	<0.001
Left superior parietal lobule	−32	−46	48	10.63	2598	<0.001
Right precentral gyrus	46	6	28	5.37	622	<0.001
Contrast “analytical > rest”						
Right inferior occipital gyrus	34	−94	0	13.80	1475	<0.001
Left precentral gyrus	−44	6	30	13.77	15,512	<0.001
Left inferior occipital gyrus	−26	−96	−2	13.55	2259	<0.001
Right cerebellum exterior	24	−70	−50	10.47	859	<0.001
Left superior parietal lobule	−28	−48	50	9.50	3006	<0.001
Right superior parietal lobule	30	−60	36	8.69	1951	<0.001
Right precentral gyrus	46	2	28	6.00	1019	<0.001
Contrast “insight > analytical”						
Right precentral gyrus	40	−20	54	5.49995	259	0.017

x, y, z—peak coordinates in the Montreal Neurological Institute (MNI) space.

**Table 2 behavsci-10-00170-t002:** Areas that showed significant effects in the meta-analysis [12], which were used as ROIs in the present study.

ROI #	Gyrus/Region	Volume (mm^3^)	Weighted Center, MNI Coordinates (mm)		
			*x*	*y*	*z*
1	Left premotor cortex	2112	−44.58	3.96	29.38
2	Left middle temporal gyrus, left precuneus	1728	−28.52	−65.42	31.45
3	Right superior frontal gyrus, left cingulate gyrus	1608	3.59	14.64	45.72
4	Left claustrum	1384	−33.41	17.68	−2.27
5	Left middle temporal gyrus, left middle occipital gyrus	976	−49.05	−58.3	−3.19
6	Left uvula	440	−5.33	−79.75	−32.55
7	Left frontal eye field	440	−27.08	−0.93	55.99
8	Right insula	368	39.29	7.35	13.77
9	Left insula	328	−39.29	12.46	10.63
10	Right precuneus	328	26.97	−69.03	47.44
11	Right middle temporal gyrus	312	52.13	−56.13	−9.21

**Table 3 behavsci-10-00170-t003:** Tests of the mean effects in the areas from the meta-analysis [12] (Table 2) for the three studied contrasts.

ROI #	Gyrus/Region	Insight > Rest		Analytical > Rest		Insight > Analytical	
		*t* (28)	qFDR	*t* (28)	qFDR	*t* (28)	qFDR
1	Left premotor cortex	12.7	**<0.001**	14.1	**<0.001**	−1.04	0.93
2	Left middle temporal gyrus, left precuneus	5.33	**<0.001**	5.89	**<0.001**	−1.51	0.93
3	Right superior frontal gyrus, left cingulate gyrus	7.31	**<0.001**	8.4	**<0.001**	−1.24	0.93
4	Left claustrum	6.86	**<0.001**	7.27	**<0.001**	−1.47	0.93
5	Left middle temporal gyrus, left middle occipital gyrus	2.85	**0.006**	3.14	**0.003**	−1.04	0.93
6	Left uvula	0.667	0.28	0.303	0.41	0.4	0.93
7	Left frontal eye field	9.79	**<0.001**	9.22	**<0.001**	−0.876	0.93
8	Right insula	−0.248	0.6	0.228	0.41	−0.698	0.93
9	Left insula	5.12	**<0.001**	6.24	**<0.001**	−0.462	0.93
10	Right precuneus	3.5	**0.001**	3.88	**<0.001**	−1.31	0.93
11	Right middle temporal gyrus	1.79	0.051	2.08	**0.028**	−1.25	0.93

t—Student’s statistic, qFDR—results of the Benjamini—Hochberg FDR adjustment applied to the *p*-values obtained in the 11 tests. Significant test results are shown in bold.

**Table 4 behavsci-10-00170-t004:** Significant correlations with the insight rate in the seed-to-voxel analysis. The seed regions and their numbering were taken from the results of the meta-analysis [12] (Table 2).

Seed Area		Connected Area						Sign of Correlation Between Connectivity and Insight Rate
ROI #	Gyrus/Region	Gyrus/Region	Volume (mm^3^)	Peak MNI Coordinates (mm)			*p*(FWE)	
				x	y	z		
1	Left premotor cortex (Figure 6A)	Right superior parietal lobule (Figure 6B)	2784	14	−46	74	0.002	+
		Right superior frontal gyrus (Figure 6B)	1928	24	−04	70	0.02	+
		Right postcentral gyrus (Figure 6B)	1808	40	−32	62	0.03	+
4	Left claustrum (Figure 7A)	Right superior and middle frontal gyri (Figure 7B)	1552	30	56	26	0.04	+
7	Left frontal eye field (Figure 8A)	Left lateral occipital cortex (Figure 8B)	2056	−30	−74	10	0.009	−

x, y, z—peak coordinates in the Montreal Neurological Institute (MNI) space.

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
