# Peer review of "Brain Activations and Functional Connectivity Patterns Associated with Insight-Based and Analytical Anagram Solving"

_behavsci, 2020, doi:10.3390/bs10110170_

Round 1
Reviewer 1 Report
The article concerns the research on the phenomenon of insight, important for understanding of the neural bases of creativity. Based on the task fmRI and resting-stage fMRI research, the authors analyzed regions of the brain with significant activations during insight-based and analytical anagram solving as well as functional connectivity associated with the rate of insight solutions. The results could be compared with the tests of the mean effects in the areas from the meta-analysis of insight-related fMRI studies by Sprugnoli et al.
The substantive discussion of the findings obtained and the resulting suggestion for further research are noteworthy. In my opinion, the work deserves to be published.
Author Response
We thank the reviewer for the careful and thorough reading of the manuscript.
Reviewer 2 Report
The paper reports a study on the neural correlates of insight and analytic problem solving. This relates to a broader research program on the brain bases of creativity. The experiment involves an anagram task, in which participants self report on how they solve each puzzle (through insight or an analytic strategy). The primary contrast of interest is between those areas activated during insight and analytic problem solving.
The work is limited by the vagueness and elusiveness of the latent variables under study (creativity and insight). The reliance on self report and the mismatch between the temporal resolution of fMRI and the "stroke of insight" make this design insensitive to detecting the neural signature of insight as it has been operationally defined.
A preferable design would have involved a manipulation. If it is difficult to design a condition where insight occurs reliably, it may be possible to contrast with a condition designed to make insight very unlikely. While the authors note that the self report is likely the optimal approach to determining whether insight has occurred in a paradigm suitable for fMRI, and this may be true, I am not convinced that this design is sensitive to what it aims to detect.
This is in part because "insight" and "analytic" are not operationally defined in the introduction. The only definition is on line 45: "Insight is contrasted with two other ways of problem solving---analytical and memory retrieval---and manifests as a sudden and unexpected discovery of the correct solution, often preceded by long unproductive attempts to find it". Does this mean that long, fruitless period of analytic reasoning following by the rapid discovery of a solution is insight (meaning that the two problem solving approaches are largely the same, but different only in the instant of the solution)? Is the difference whether the solution is arrived at through deduction or induction? Does insight follow a search, while an analytic solution comes from following rules? If this is the case, then how can an anagram task sometimes be analytic and other times be insight? This is an important part of the motivation for the design, but it is not discussed well in the introduction.
Referring to both the 9 dot problem and the 8 coin problem as insight tasks only furthers my confusion. The former is a playground for lateral thinking, and the latter is an algorithmic puzzle with a optimal answer: weigh 3 and 3. If they balance you got it in one. If they don't, compare two coins from the lighter scale. Done. The authors note that the variety of tasks purporting to study insight is a problem, and I completely agree. "Insight" lacks a clear definition. I appreciate this is all established in the literature, and these tasks are both used to study "insight". But comparing and contrasting these tasks does not clarify the definition of insight for me.
The discussion section is good and realistic about what can be concluded and what the limitations are. The literature review appears thorough when describing what parts of the brain have been identified in prior work and what functions are thought to be associated. However, given that there is very little difference between insight and analytic conditions (and that which does exist is noted to likely be associated with a motor response confound), it seems that this work largely replicates prior work, but with a task that does not seem to differentiate between insightful and analytic problem solving. Because the experimental design does not differentiate these processes, there is neither a novel empirical result nor a clear methodological contribution.
The recommendation of using combined EEG/fMRI in the conclusion of the paper is interesting. However, the recommendation is motivated by the observation (assumption?) that insight is a fleeting moment. Regardless of whether this is true or not, the current work does not actually evaluate hypotheses about the timecourse of insight. The recommendation might have been made without running the experiment at all. The authors due observe that the problems particpants report solving through insight are solved more quickly than problems reportedly solved through an analytic process, which is consistent with prior work. Yet this does not speak to the similarity or difference of the processing the leads up to a solution on insight or analytic trials. The null contrast between insight and analysis trials is consistent with this recommendation, but does not provide evidence or support inference about the fleeting nature of insight. Indeed, all of my confusion about how to define insight remains after consuming the results of this study.
Many of the critiques I have recorded above are acknowledged by the authors themselves. While I feel that this study was well executed and the analyses are technically sound, I am reluctant to draw conclusions based on the work given the limitations of fMRI relative to the focus of study, the unfortunate reliance on self report grouping rather than an experimental manipulation, and inherent challenges of studying insight (a phenomenon which seems to lack a useful operational definition).
That said, the study clearly replicates prior work with no surprises, and this adds data to a field of study that needs more. Although, as the authors note (line 328): "it can be hypothesized that most of the activation areas that we identified are associated with the process of solving the task and not with insight per se. This is also indirectly suggested by the overlap of the identified activation areas with the brain regions involved in divergent thinking, general cognitive control and attention, and belonging to several brain networks (default mode network (DMN), salience, executive control, and creativity networks)." In short, while these are areas relevant to problem solving, it's not possible to say that any of what has been observed is specific to insight.
Minor issues:
- The reporting of the rs-fMRI is hard to follow. When referred to in text, it is not clear what the seed is and what the terminus is, and splitting seed and terminus across multiple figures is also hard to follow.
- Line 77: "of different brain structures". This is too vague.
- Line 79: need citations for the saliency and creativity networks.
- Table 1: p-values are not effect sizes--this table is incomplete. Poldrack et al (2008; Neuroimage) recommend that the "Minimum data to be included in a table should include location of activation in stereotactic space (e.g., that of the maximum for voxel-level inference), statistics regarding the activation cluster (including maximum statistic value and size of the cluster), and anatomical labels. The means by which the anatomical labels are derived (e.g., an atlas or automated labeling method) should be clearly specified. We also recommend that tables or figures include some form of effect size measure (e.g., mean percent signal change and standard deviation) in order to allow future meta-analyses."
- Figure 5: is this insight > analytic, or vice versa?
Reviewer 3 Report
The authors aim at distinguishing neural correlates for anagram solutions by insight or an unspecified and hence distinguishable analytical strategy (probably only by lack of AHA experience). Furthermore, they can not sufficiently rule out a potential confound regarding their main finding, i.e., differences in brain activation depending on type of solution. The paper is written for an expert readership familiar with the literature on insight and neuronal correlates and correctly identifies several drawbacks when investigating insight using fMRI but still employs this method and does not use proper experimental design (i.e., counterbalancing of responses to be given). Overall, the results do not come as a surprise mainly because the analytical solution is never properly defined (i.e., by means of response pattern, error rates to lures) or operationalised and therefore no specific pattern is expected (or at least not lined out by the authors). Finally, I wondered whether the design using anagrams twice for the selection task and the fMRI paradigm is not risky in that you might measures strategies participants adopt in the selection task in the fMRI part of the study given those worked well and hence insight rates reported should be tested to be significantly different from chance level (50%) and reported. Overall, I have to state that I was not convinced by the study design and the missing experimental control prevents me from writing a more positive review. Finally, I was surprised by the consistent value reported in Table 1 for analytical vs. insight contrast.
Round 2
Reviewer 2 Report
While I appreciate the care and consideration taken by the authors in responding to my concerns, my reservations about the work remain: the distinction between insight and analytic thinking is not clear to me, even with the additional definitions and attempts at clarification. For example, part of the cited definition of insight is that it "is often preceded by long unproductive attempts to find a solution", which appears at odds with the finding that responses were provided more quickly on trials where participants reported insight.
I appreciate that the exact instructions given to participants about how to determine whether they have experienced "insight" are not provided. But: how many of the decisions and actions that people take can actually be described the steps we take to arrive at them? Are we to consider all of these insight?
How is an anagram solved analytically? Yes, we are sensitive to bi-gram frequency, which may help reduce the dimensionality of the problem; yes, we are sensitive to the fact that some letters are likely to appear at the beginning middle or end of a word. These factors could be used analytically, but distributional statistics affect behavior without the need to reason about them. Is a lexical decision task analytic, or driven by insight? There are already 120 permutations of five letters, and 40,320 permutations of eight. What is the analytic approach to solving an anagram?
The authors write that classic insight tasks "require restructuring their representations and abandoning the usual algorithms or solutions and probably cannot be solved through other problem-solving strategies (i.e., the analytical method; Sprugnoli et al., 2017)". This statement makes it difficult to understand how the anagram task can be both on different trials, such that it is a function of the participant and not the stimulus that determines whether it is insight or not.
It appears that prior work has presented participants with rules or semantic category cues to promote a more consistent strategy. Especially with the rule-driven approach (e.g., switch the first and the last and the middle two: HANWDICS --> SANDWICH; there could be more complex rules of course), this would be obviously analytic. Trials without rules might be solved analytically or by insight, but trials with rules would have an analytic bias, I presume.
Anyway, I trust that my concerns about the constructs under study are well appreciated at this point, and I'll drop it. It appears the authors are in line with the literature they are contributing to. They have clarified their definitions of insight and analytic, and I'm not going to hold up the paper on account of these issues.
However, I do have a number of minor revisions that are necessary before this paper can be published, primarily with respect to the reporting that statistics and neuroimaging results.
Minor comments:
- The "t" in "t-test" or "t-statistic" is lowercase by convention.
- Report degrees of freedom for every paramatric statistical test.
- Report the actual test-statistic for your Wilcoxon test, not just the p-value.
- It is not conventional to report p-values in scientific notation. APA style recommends reporting exact p-values when p >= .001, and just p < .001 otherwise. A p-value is not an effect size, so it doesn't matter beyond facilitating your null hypothesis test.
- Table 2 does not satisfy the "minimum requirements" proposed by Poldrack et al (2008; Neuroimage): "Minimum data to be included in a table should include location of activation in stereotactic space (e.g., that of the maximum for voxel-level inference), statistics regarding the activation cluster (including maximum statistic value and size of the cluster), and anatomical labels. The means by which the anatomical labels are derived (e.g., an atlas or automated labeling method) should be clearly specified. We also recommend that tables or figures include some form of effect size measure (e.g., mean percent signal change and standard deviation) in order to allow future meta-analyses."
--- Even though you report the meta analysis peak coordinates in Table 1, it is still important that you report the full data for your own analyses.
--- While on the topic of tables: the over-use of border lines makes the table hard to read. APA style recommends a cleaner table format.
Reviewer 3 Report
I have to thank the authors for their work invested. However, my main concern about the reported difference between insight and analytical anagram solution was not assessed at it is still valid that the response mapping might have caused the difference as the authors correctly state. I can see the point of not running a control experiments as scanning is hard these days given the pandemia and am therefore satisfied with clarification made by the authors - congratulations to this nice piece of work
Author Response
We thank the reviewer for the attention to this study and the revision suggestions. We hope that we have been able to address them, improving the quality of the result presentation in the paper.